# Fate mapping reveals mixed embryonic origin and unique developmental codes of mouse forebrain septal neurons

Lorenza Magno [1✉], Zeinab Asgarian[1], Migle Apanaviciute[1], Yasmin Milner[1], Nora Bengoa-Vergniory[1], Anna Noren Rubin[1] & Nicoletta Kessaris [1✉]

The septum is a key structure at the core of the forebrain that integrates inputs and relays information to other brain areas to support cognition and behaviours such as feeding and locomotion. Underlying these functions is a rich diversity of neuronal types and an intricate complexity of wiring across and within the septal region. We currently have very little understanding of how septal neuronal diversity emerges during development. Using transgenic mice expressing Cre in different subsets of telencephalic precursors we explored the origins of the three main neuronal types of the septal complex: GABAergic, cholinergic and glutamatergic neurons. We find that septal neurons originate from distinct neuroepithelial domains of the developing septum and are born at different embryonic time points. An exception to this is the GABAergic medial septal Parvalbumin-expressing population which is generated outside the septum from surrounding germinal zones. We identify the transcription factor BSX as being expressed in the developing glutamatergic neuron population. Embryonic elimination of BSX in the septum results in a reduction of septal glutamatergic cell numbers and a consequent deficit in locomotion. Further refinement of septal neuron diversity is needed to understand the multiple roles of septal neurons and their contribution to distinct behaviours.

[1] Wolfson Institute for Biomedical Research and Department of Cell and Developmental Biology, University College London, WC1E 6BT London, UK.
✉email: l.magno@ucl.ac.uk; n.kessaris@ucl.ac.uk

The septal complex is an expanding structure in primate evolution that forms an integral part of the limbic system, connecting the telencephalon with the hypothalamus and brain stem[1–3]. Although largely considered a relay centre, its reciprocal connections with the hippocampus and rhythmic drive of hippocampal theta demonstrate a prominent role in cognitive processing and coordination across different nuclei. In addition, the progressive deterioration of septal and extra-septal cholinergic neurons of the basal forebrain in early Alzheimer's disease patients and during normal aging further implicates this structure in cognition and memory[4–6].

The septo-hippocampal projection system is the largest component of the medial septum (MS) and the ventral limb of the diagonal band (vDB) region. It consists of theta pace-maker GABAergic, and modulatory cholinergic ascending projections[7–12], as well as a more recently-identified glutama-tergic MS projection to the hippocampus[13–15]. Back-projections from the hippocampus targeting MS and lateral septal (LS) components also participate in theta rhythmicity[16,17]. In addition to hippocampal and cortical projections, which convey cognitive information, the LS receives affective information from the amygdala, hypothalamus and bed nucleus of the stria terminalis, acting as a nodal point for information relay to diencephalic, mesencephalic and rhomboencephalic regions[3,18]. The LS is almost entirely GABAergic but is characterised by expression of a wide range of markers, including calcium-binding proteins, peptides and hormones[19].

In addition to cognitive processing, the septum participates in the organisation and execution of voluntary motor behaviours through a process of sensorimotor integration. MS lesions and inactivation of the MS both result in reduced locomotion and motor activity[20,21]. More recent studies further support the involvement of hippocampal theta in voluntary locomotion in rodents and humans[22–24]. In particular, MSvDB glutamatergic neurons projecting to the hippocampus control the initiation, speed and duration of locomotion, as well as the entrainment of hippocampal theta oscillations before locomotion onset[24]. The LS is thought to mediate the regularity of theta oscillations and locomotion and integrate locomotion and reinforcement beha-viour through its descending feedback from the hippocampus and connections to the lateral hypothalamus and ventral tegmental area[25].

We know little about how different septal neurons are specified and how they assemble into circuits. A few studies that addressed the development of this region provide evidence for a temporal and spatial bias in the generation of the various neuronal types populating the different septal nuclei[26–30]. The septal embryonic progenitor domain has been broadly subdivided into pallial and subpallial regions[31,32] and other smaller subdomains according to the expression of molecular markers[33]. The contribution of these regions to different septal neurons and the importance of the molecular subdivisions of the septal neuroepithelium remain poorly characterised.

We used a series of transgenic mice expressing Cre recombi-nase in different forebrain progenitor regions to identify the origin and lineage of neuronal populations in the adult septum. We focused on nuclei of the MSvDB and subpopulations of LS neurons. We find that most septal neurons examined and, in particular, most LS neurons, are generated from different focal zones within the septal neuroepithelium. On the other hand, Parvalbumin (PV)-expressing neurons of the MSvDB originate from surrounding neuroepithelial zones. MSvDB glutamatergic neurons have a largely septal origin and can be identified by expression of unique genetic markers. Using mice expressing Cre under control of the homeobox-encoding gene *Bsx*, as well as *Bsx* conditional loss-of-function mice, we demonstrate that this transcription factor is required for the development of MSvDB glutamatergic neurons and their contribution to mouse loco-motor activity.

## Results

**Neurons of the septum.** At a gross anatomical level, the boundaries of the MSvDB and the LS can be identified by their different cytoarchitecture; the LS has a complex organisation of subnuclei while the MS is arranged in layers or lamellae of neu-rons around an axis of symmetry[34–36]. We examined the LS and MSvDB for expression of neurotransmitter markers as well as the three calcium-binding proteins Calbindin (CB), Calretinin (CR) and Parvalbumin (PV), all of which had been reported in the septum in previous studies (Fig. 1). We focused on three different Bregma levels that we refer to as Rostral (R), Intermediate (I) and Caudal (C) (Fig. 1f, see also Methods section). Quantification of absolute and average number of marker[+ve] cells per level is shown in Table 1. Where possible, we birth-dated these neurons in order to determine their temporal emergence during embry-ogenesis (Supplementary Fig. 1).

CB-, CR- and PV-expressing GABAergic neurons have previously been identified in the LS[37,38]. We detected CB- and CR-expressing cells at all levels of the LS and these formed largely non-overlapping populations (<5% of CB[+ve] cells are double-labelled for CR and <8% of CR[+ve] cells co-label with CB - tested using two different antibodies per marker (see Methods section; Fig. 1a–c, g, h). Small numbers of PV-expressing neurons are found scattered throughout the LS, and these do not overlap with either CB or CR (Fig. 1a, b, g, h). Birth-dating LS populations at 2-day intervals from E10.5 to E18.5 using EdU incorporation every other day and postnatal analysis at P30, showed a peak of PV neuron generation at E10.5, whereas all other populations examined peaked at later stages (Supple-mentary Fig. 1a, b).

The MS is continuous ventrally with the vDB and, for the purpose of this study, we analysed the entire MSvDB complex. GABAergic PV-expressing projection neurons constitute the core of the MS laminar structure, while cholinergic neurons - identified by expression of p75[NTR] - occupy a more lateral position (Fig. 1a, d, g, i). PV and p75[NTR] neurons form entirely distinct populations (Fig. 1g, h)[34]. CB and CR-immunoreactive neurons also constitute a large portion of this region, partially intermingling with cholinergic neurons, but also extending to a more lateral position, bordering with the lateral septum[35] (Fig. 1a, d, g). A small overlap between CB and CR expression exists (<15% of CB are double-labelled for CR and <15% of CR co-label with CB) (Fig. 1a, d, e, h). <10% overlap exists between PV and CR and between CB and p75[NTR]. Birth-dating these neurons during embryogenesis showed early emergence of PV[+ve] and CR[+ve] populations and a later generation of p75[NTR] and CB[+ve] neurons (Supplementary Fig. 1a, c).

In addition to the GABAergic and cholinergic cells in the MSvDB and the LS, a population of neurons that uses glutamate as a neurotransmitter has been identified[13–15,39]. These express the vesicular glutamate transporter 2 (VG2)[40]. VG2[+ve] MSvDB neurons display a heterogeneous firing pattern. They provide local excitatory input to cholinergic and GABAergic septal neurons[41], as well as long-range input to the hippocampus via the fornix. VG2[+ve] neurons are found mainly within the MSvDB, intermingled among cholinergic neurons (Fig. 1i). A small population of VG2[+ve] neurons is also scattered within the LS. GABAergic, cholinergic and glutamatergic neurons of the MSvDB are largely non-overlapping populations with only $1.4 \pm 0.9\%$ of the cholinergic and $1.6 \pm 0.2\%$ of the glutamatergic neurons co-labelling with *GAD1* (Fig. 1i–m).

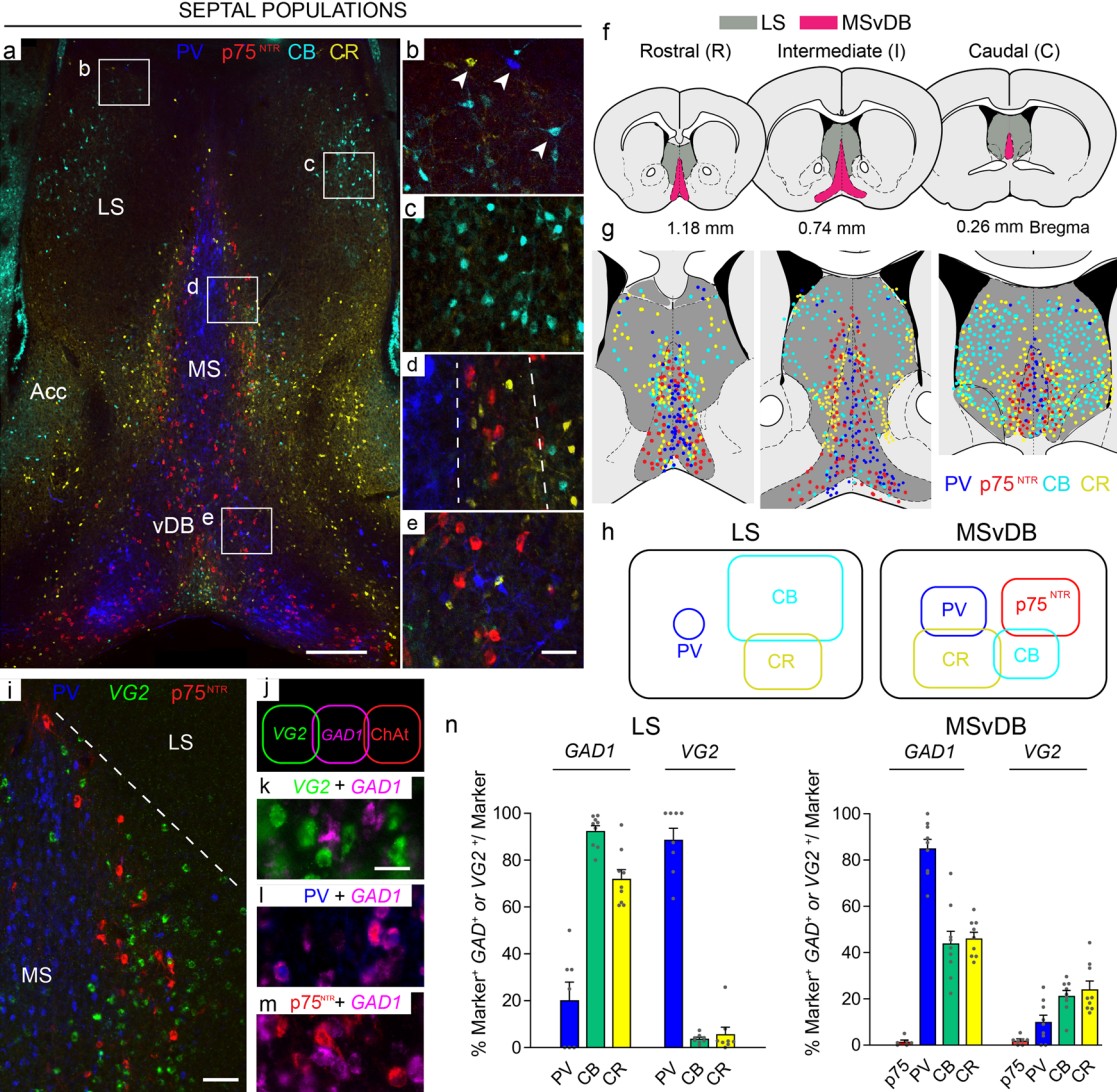

**Fig. 1 Chemoarchitecture of the septal complex. a** Representative coronal sections through the adult mouse brain showing the organisation of septal nuclei. The four main neuronal subpopulations examined in this study are: PV (blue), p75[NTR] (red), CR (yellow) and CB (cyan). The picture was obtained by merging images from two consecutive sections. **b–e** Boxed areas in **a** are shown at higher magnification: **b**, **c** LS; **d** MS; **e** vDB. **b** Immunoreactivity for the three calcium-binding proteins in non-overlapping neuronal populations throughout the LS (arrowheads). **c** CB-expressing neurons in the LS show the typical arrangement of a nuclear structure. **d** The laminar organisation of the MS is shown, dashed lines indicate the borders of the layers. **e** p75[NTR] -, CB- and PV-immunoreactive neuronal populations are interspersed at the border of the MS and vDB. **f** Diagrams showing the three rostro-caudal levels analysed in this study. MSvDB and LS boundaries used in counts are shown (modified from ref. [71]) and distribution of septal populations at rostral (R), intermediate (I) and caudal (C) levels. **g** Diagram showing relative abundance and overlap among the different populations of septal neurons at the three different levels examined in this study **h** Summary of the neuronal markers examined in this study and their overlap in the LS and the MSvDB. All overlaps indicated are <15%. **i** GABAergic (PV-expressing), glutamatergic and cholinergic neurons in the MSvDB. **j** MSvDB neurons expressing different neurotransmitters represent largely distinct populations. **k–m** Representative overlap between GABAergic, glutamatergic and cholinergic populations. **n** Quantification of co-localisation between *GAD1* or *VG2* cells and markers in the LS and MSvDB. $n = 3$ independent mice used for each marker. LS: GAD1 PV $n = 7$ sections; CB, CR $n = 9$ sections. VG2 PV, CB, CR $n = 8$ sections. MSvDB GAD1 p75 $n = 6$ sections; PV, CB, CR $n = 9$ sections. VG2 p75[NTR] $n = 7$ sections; PV, CB, CR $n = 9$ sections. Histograms show mean + SEM. Source data are provided in Supplementary Data 1. LS lateral septum, MS medial septum, Acc shell of the nucleus accumbens, vDB vertical limb of the diagonal band. Scale bars: **a** 200 μm, **e**, **i** 50 μm, **k** 10 μm.

We examined the neurotransmitter phenotype of septal p75[NTR], PV, CB and CR neurons by quantifying their co-expression of *GAD1* (*Gad67*) and *VG2*, markers that detect inhibitory and excitatory neuronal phenotypes, respectively. In the LS, the majority of CB[+ve] and CR[+ve] cells co-express *GAD1* (92.5 ± 2.3%, and 70.3 ± 4.7%, respectively), whereas nearly all LS PV[+ve] neurons express *VG2* (94.2 ± 3.3%; Fig. 1n). In contrast, PV[+ve] neurons in the MSvDB are GABAergic (95.8 ± 4.2%), as are some CB[+ve] and CR cells[+ve] (43.9 % ± 5.2%, and 46.12 ± 2.6%, respectively). Small numbers of PV[+ve] (7.6 ± 3.3%), CB[+ve] (21.3 ± 2.3%) and CR[+ve]

(26.17 ± 3.8) MSvDB neurons co-express *VG2* (Fig. 1n). Only 1.1 ± 0.5% of the p75[NTR] immunolabelled neurons co-express *VG2* and these constitute 0.6 ± 0.3% of the *VG2* MSvDB population.

In summary, the septal region is populated by a large variety of neuronal subtypes, which show characteristic marker expression and distribution. The distribution of EdU-incorporating cells within the septum shows an outside-in generation of septal nuclei with respect to the ventricular zones and the telencephalic lumen: early-born cells are located medially within the septum whereas late-born cells occupy lateral positions close to the lumen.

**Table 1 Cells counted in the MSvDB and the LS.**

| Marker | Level | Mouse lines | Animals | Total cells counted | Average number of cells | Std | SEM |
|---|---|---|---|---|---|---|---|
| **MSvDB** | | | | | | | |
| p75$^{NTR}$ | R | 8 | 24 ($n = 3$ each line) | 1649 | 69 | 25.41 | 5.19 |
| | I | 8 | 24 ($n = 3$ each line) | 3691 | 153.79 | 21.06 | 4.30 |
| | C | 8 | 24 ($n = 3$ each line) | 1446 | 60.25 | 10.91 | 2.23 |
| CB | R | 8 | 24 (n=3 each line) | 1836 | 77 | 18.23 | 3.72 |
| | I | 8 | 24 ($n = 3$ each line) | 2627 | 109 | 25.62 | 5.23 |
| | C | 8 | 24 ($n = 3$ each line) | 2500 | 105 | 19.41 | 3.96 |
| PV | R | 8 | 24 ($n = 3$ each line) | 850 | 39 | 31.57 | 6.44 |
| | I | 8 | 24 ($n = 3$ each line) | 2429 | 101 | 22.92 | 4.68 |
| | C | 8 | 24 ($n = 3$ each line) | 1350 | 56 | 16.03 | 3.27 |
| CR | R | 8 | 24 ($n = 3$ each line) | 1859 | 81 | 23.21 | 4.74 |
| | I | 8 | 24 ($n = 3$ each line) | 3729 | 155 | 24.62 | 5.03 |
| | C | 8 | 24 ($n = 3$ each line) | 2198 | 96 | 20.48 | 4.18 |
| *VG2* | R | 7 | 21 ($n = 3$ each line) | 2444 | 116 | 29.75 | 6.49 |
| | I | 7 | 21 ($n = 3$ each line) | 3982 | 182 | 51.26 | 11.19 |
| | C | 7 | 21 ($n = 3$ each line) | 5121 | 254 | 79.25 | 17.29 |
| **LS** | | | | | | | |
| CB | R | 8 | 24 ($n = 3$ each line) | 1070 | 24 | 11.64 | 2.38 |
| | I | 8 | 24 ($n = 3$ each line) | 4124 | 172 | 98 | 20.04 |
| | C | 8 | 24 ($n = 3$ each line) | 17,550 | 366 | 158 | 32.22 |
| PV | R | 8 | 24 ($n = 3$ each line) | 239 | 6 | 5 | 1.03 |
| | I | 8 | 24 ($n = 3$ each line) | 290 | 12 | 5 | 1.02 |
| | C | 8 | 24 ($n = 3$ each line) | 335 | 8 | 10 | 2.08 |
| CR | R | 8 | 24 ($n = 3$ each line) | 1421 | 32 | 30 | 6.06 |
| | I | 8 | 24 ($n = 3$ each line) | 2417 | 101 | 37.01 | 7.55 |
| | C | 8 | 24 ($n = 3$ each line) | 7242 | 161 | 182 | 37.18 |
| *VG2* | R | 7 | 21 ($n = 3$ each line) | 540 | 26 | 15 | 3.27 |
| | I | 7 | 21 ($n = 3$ each line) | 695 | 35 | 23 | 5.05 |
| | C | 7 | 21 ($n = 3$ each line) | 991 | 52 | 30 | 6.46 |

Cells expressing p75$^{NTR}$, CB, PV, CR and *VG2* were counted at three different rostro-caudal levels (R, I, C) of the MSvDB and the LS for lineage tracing purposes. The number of mouse lines as well as the total number of animals used for each marker are shown. The total number of marker$^{+ve}$ cells counted for each of these markers as well as the average number of cells counted per animal are also indicated for each level.

MSvDB PV neurons, which occupy the most medial position of the septum, constitute one of the first populations of neurons to be generated.

**Septal and extra-septal neuroepithelial zones generate LS and MSvDB neurons.** In order to identify the embryonic origin of septal neurons we made use of a series of Cre-expressing transgenic mouse lines crossed to suitable reporters to label septal and extra-septal neuroepithelial cells (Fig. 2). Emx1-Cre;GFP labels the pallial neuroepithelium[42] (Fig. 2a, b). Zic4-Cre;YFP labels the entire septal neuroepithelium at all dorso-ventral and anterior-posterior levels and is mutually exclusive with the pallial Emx1-Cre[43,44] (Fig. 2a, b). Nkx2.1-Cre;YFP labels a caudo-ventral domain of the septal neuroepithelium, as well as the medial ganglionic eminence (MGE) and the preoptic area (POA) and overlaps partly with Zic4-Cre[42,44,45]. Shh-Cre;YFP labels a subdivision of the POA and a small region of the ventral MGE[46,47] (Fig. 2a, b). We also labelled embryonic neurons within the SVZ/mantle of the telencephalon using mice expressing Cre under control of *Lhx6*[45] (Fig. 2a, c), *Dbx1*[48], and two newly-generated mouse lines expressing Cre under control of *Lhx7* and *Bsx*, respectively (Fig. 2a, c and Supplementary Fig. 2). *Lhx7* is known to be expressed in forebrain cholinergic neurons, whereas the expression of *Bsx* in the septum is uncharacterised[49]. A summary of the Cre-expressing lines used in our study is shown in Fig. 2d, e.

In order to fate-map the embryonic origin of septal neurons, we quantified the extent of co-localisation between the reporter gene (GFP, YFP or tdTomato) and the various septal neuronal markers in Cre-expressing mice at P30 at the three rostro-caudal levels

shown in Fig. 1f. We found little or no contribution of the pallial neuroepithelium to LS and MSvDB neurons using Emx1-Cre;GFP mice (Figs. 3 and 4), with the exception of small numbers of LS CR neurons where <5% of the population expressed GFP (Fig. 3a, c). The vast majority of LS CR-expressing cells have a dorsal septal origin (as evidenced by reduced labelling in Nkx2-1Cre;YFP mice), while CB and PV LS populations have a mixed dorsal and ventral septal origin (Fig. 3a–d). Only small subsets of CB$^{+ve}$, CR$^{+ve}$ and PV$^{+ve}$ neurons were labelled with Lhx6-Cre, whereas Dbx1-Cre and Bsx-Cre made a bigger contribution to these populations (Fig. 3a–d). Lhx7-Cre labelled subsets of CB$^{+ve}$ cells and was excluded from CR$^{+ve}$ and PV$^{+ve}$ populations. Given the absence of p75$^{NTR+ve}$ cholinergic neurons in the LS and the known role for LHX7 in the determination of cholinergic versus GABAergic cell fate[50], these CB cells in the LS may represent non-cholinergic cells that may have expressed low levels of the Cre transgene but never became cholinergic. Overall, all LS neurons examined have a predominantly septal origin, with the majority of neurons being labelled with YFP in Zic4-Cre;YFP mice (Fig. 3a–d).

p75$^{NTR+ve}$ cholinergic MSvDB neurons at all three rostro-caudal levels examined originate exclusively from *Nkx2.1*-expressing neuroepithelial cells in the caudo-ventral embryonic septum (Fig. 4a, b), as recently shown[43]. In contrast, PV-expressing cells of the MSvDB have a mixed origin that lies outside the septum, as evidenced by minimal labelling of this population in Zic4-Cre;YFP mice (Fig. 4a, c). Partial labelling of this population in Nkx2.1-Cre;YFP and Shh-Cre;YFP mice suggests that these cells may have a dual MGE/POA origin (Fig. 4c). Successful labelling of nearly all MSvDB PV neurons in Nestin-Cre;YFP mice - which should label all neuroepithelial

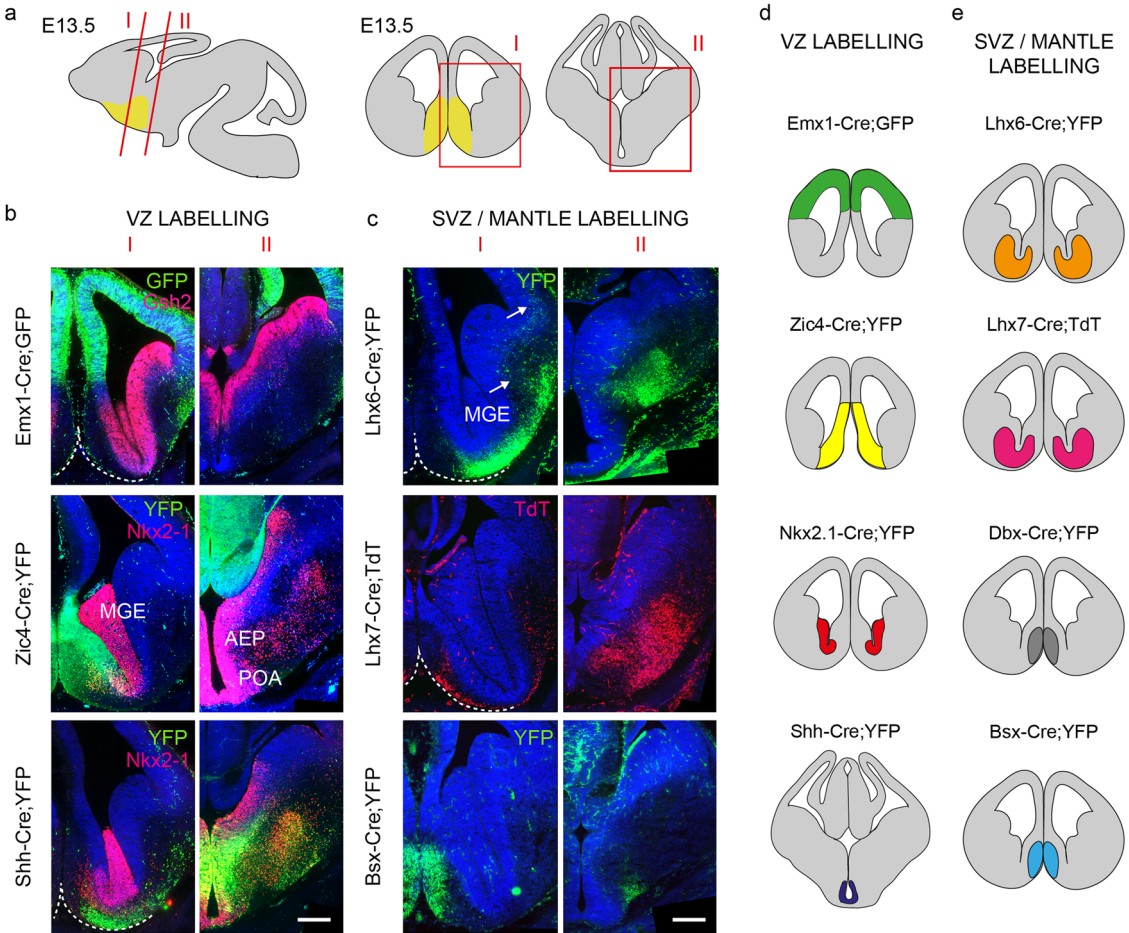

**Fig. 2 Transgenic mice used to fate-map the embryonic septum. a** Schematic representation of a sagittal E13.5 mouse brain section and the corresponding coronal cuts shown in I and II. Boxed areas in red are shown for rostral (I) and caudal (II) levels. The position of the septum is indicated in yellow. **b** YFP/GFP expression in forebrain germinal regions (ventricular zone, VZ) of three lines used for fate mapping. Cre recombination can be detected in the neocortex (Emx1-Cre;GFP), the septum (Zic4-Cre;YFP) and the AEP/POA (Shh-Cre;YFP). Immunolabelling for Gsh2 and Nkx-2-1 delineates the subpallium and the MGE/POA regions, respectively. Scale bar: 200 μm. **c** Transgenic lines with expression in the subventricular (SVZ) and mantle zones. In Lhx6-Cre;YFP, YFP labelling spans the MGE SVZ and mantle. YFP+ve neurons initiating their migration toward the cortex are indicated with white arrows. TdT expression is detected in presumptive immature cholinergic forebrain neurons in Lhx7-Cre;TdT. YFP immunoreactivity can be detected in the mantle zone of the septum in Bsx-Cre;YFP mice. Ventral boundaries are indicated by a dashed white line. AEP anterior entopeduncular region, MGE medial ganglionic eminence, POA preoptic area. Scale bar: 200 μm. **d, e** Summary of VZ- and SVZ/mantle-expressing Cre lines used in this study. The area of expression for each transgenic line is identified by a different colour. The same colour coding is used in subsequent fate-mapping Figs. 3–5.

forebrain regions early during development – indicates that partial labelling in other Cre mice is not caused by a failure of reporter expression (Supplementary Fig. 3a–c). CB+ve and CR+ve MSvDB neurons originate from dorsal and ventral neuroepithelial septal precursors, as they showed almost complete co-localisation with YFP in Zic4-Cre;YFP mice, and partial co-localisation with YFP in Nkx2-1-Cre;YFP mice (Fig. 4a, d, e).

Altogether, our data show that all neurons examined, with the exception of MSvDB PV+ve cells, originate from septal neuroepithelial precursors. LS CR+ve cells have an exclusive dorsal-septal origin and MSvDB cholinergic neurons have an exclusive ventral septal origin. All other neuronal populations examined are generated from both dorsal and ventral septal precursors. MSvDB PV+ve cells originate outside the septum from surrounding precursors. A summary of the contribution of the various germinal regions and precursors to the septal populations analysed in this study is shown in Table 2.

Labelling of septal neurons with mouse lines expressing Cre in the SVZ/mantle showed that p75NTR-immunoreactive cholinergic neurons in the MSvDB originate from neural precursors/

immature neurons expressing *Lhx6* and *Lhx7* (Fig. 4a, b), as previously shown for other forebrain cholinergic neurons[50]. Comparable numbers of PV+ve neurons in the MSvDB co-expressed YFP in Lhx6-Cre;YFP and Nkx2.1-Cre;YFP mice (Fig. 4a, c), consistent with the notion that LHX6 is activated downstream of NKX2.1[51]. Variable proportions of p75NTR+ve, PV+ve, CB+ve and CR+ve MSvDB cells were labelled with YFP in Dbx1-Cre mice, suggesting heterogeneity within these populations (Fig. 4a–e). There was no co-labelling between PV or p75NTR with YFP in Bsx-Cre;YFP mice, indicating that Bsx-Cre labels a distinct population of cells within the septum (Fig. 4b, c).

**Septal origins and unique identifiers of septal glutamatergic neurons.** *VG2* transcripts can be detected in the developing medial septum at early developmental stages in a mutually exclusive pattern to *GAD1* (Fig. 5a). The location and pattern of *VG2* expression is reminiscent of *Bsx* in the embryonic septum (Fig. 2c, e). We therefore used our Bsx-Cre;YFP mice and the rest

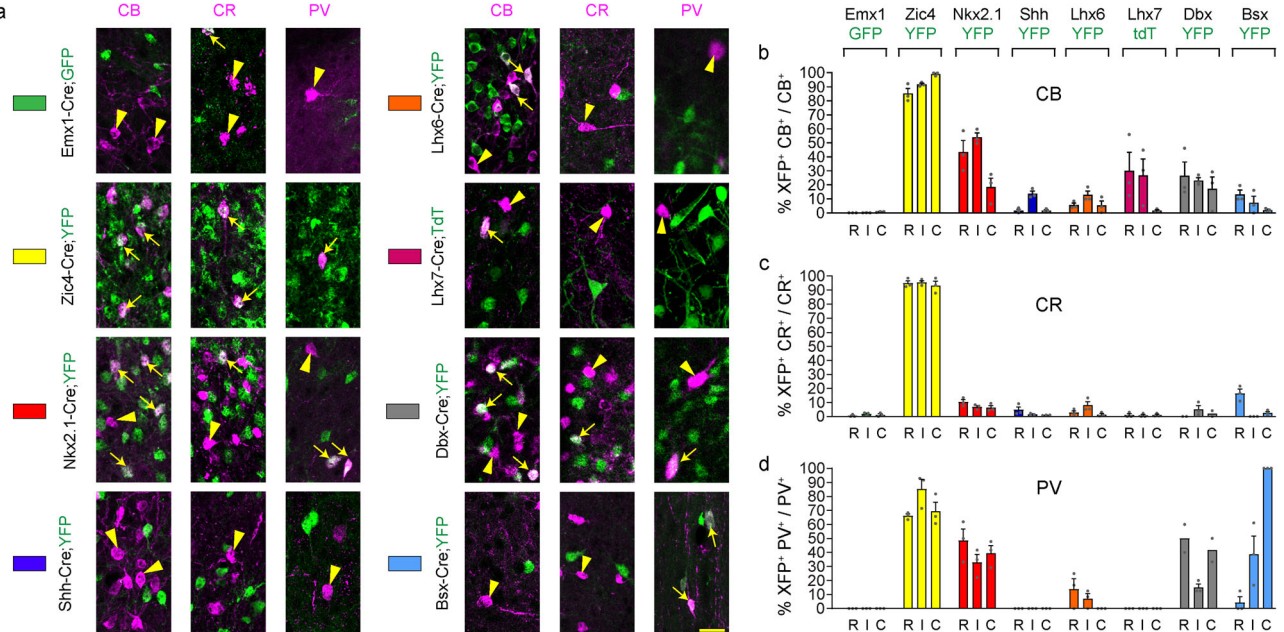

**Fig. 3 Septal embryonic origins of lateral septal neurons. a** Contribution of different forebrain domains to LS populations. Double-labelling for CB, CR or PV and the fluorescent reporter protein GFP/YFP/TdT in the various transgenic lines at P30. White arrows and arrowheads indicate double and single labelled cells, respectively. **b–d** Histograms showing the percentage of neurons double labelled for GFP/YFP/TdT over the total population. Each transgenic mouse line is identified by a different colour as shown in **a** and as summarised in Fig. 2d, e. n = 3 brains per transgenic line per marker except the following where n = 2 for Dbx-Cre;YFP mice: CR R and C, PV R and C. Histograms show mean + SEM. Source data are provided in Supplementary Data 1. Scale bar: 25 µm.

of the panel of Cre mice, to examine the origin of septal glutamatergic neurons. In the MSvDB nearly all *VG2*-expressing neurons were labelled with YFP in Bsx-Cre;YFP mice at the three Bregma levels examined (R, 92.6 ± 2.5%; I, 89.6 ± 2.3%; C, 88.7 ± 1.9%; Fig. 5b, c). The vast majority of these were also labelled in Zic4-Cre;YFP mice (R, 72.9 ± 9.4%; I, 73.2 ± 11.8%; C, 68.9 ± 3.7%; Fig. 5b, c). *Dbx1*, a transcription factor-encoding gene known for its essential role in the generation of transient forebrain glutamatergic Cajal-Retzius cells, shows a history of expression within subsets of septal glutamatergic neurons, suggesting a heterogeneity within the population and a possible role for this gene in their development (Fig. 5b, c). Small numbers of scattered glutamatergic neurons are also found in the lateral septum and these show similar origins as those in the MSvDB (Figs. 1n and 5d). In conclusion, our data show that septal *VG2* neurons originate largely from septal progenitors. Nearly all septal glutamatergic neurons are generated from precursors/immature neurons expressing *Bsx*, identifying *Bsx* as an early embryonic marker for these cells.

**A role for Bsx in septal glutamatergic neurons and locomotor behaviour.** BSX is an evolutionarily conserved homeobox-encoding gene expressed in the septum, epiphysis, mammillary bodies and arcuate nucleus[49]. In the absence of BSX, in *Bsx* germline loss-of-function mice, hypothalamic neurons display abnormal maturation and reduction of NPY[52]. To assess the requirement for BSX in septal neurons without affecting the hypothalamus, we generated mice carrying a conditional *Bsx* allele and crossed these to Zic4-Cre to obtain Zic4-Cre;Bsx^fl/fl mice (referred to as septal Bsx cKO; Fig. 6a and Supplementary Fig. 4). Within the septum, we could not directly identify the cells undergoing recombination of the *Bsx* allele due to lack of antibodies that detect the BSX protein. However, given that ~100% of VG2^+ve MSvDB neurons are labelled in Bsx-Cre mice and ~70% of the VG2^+ve population is labelled in Zic4-Cre mice (Fig. 5c),

we can infer that the majority of BSX-expressing VG2^+ve cells in this region will undergo recombination in septal BsxcKO mice. Outside the septum, we detected an overlap between YFP and *Bsx* in the epiphysis in Zic4-Cre;YFP embryos but virtually no overlap in the developing hypothalamus (Supplementary Fig. 5a). The absence of overlap between Zic4-Cre and *Bsx* in the hypothalamus was also confirmed by normal expression of *Npy* in the arcuate nucleus of adult Bsx cKO mice (Supplementary Fig. 5b).

Embryonic deletion of *Bsx* using Zic4-Cre did not result in noticeable loss of VG2^+ve cells in the embryonic septum (Fig. 6b). We quantified VG2^+ve neurons of the MSvDB in male and female adult control and Bsx cKO mice at five different anterior-posterior Bregma levels (levels 1 – 5 in anterior-posterior order; see Methods section). Significant loss of *VG2* expression was detected in both sexes in Bsx cKO mice at posterior septal levels (Fig. 6c, f). There was no significant loss of CR^+ve, PV^+ve, CB^+ve or p75^NTR+ve neurons in the MS upon deletion of *Bsx* with Zic4-Cre (Fig. 6d–f). Our data suggest that *Bsx* is required for differentiation/survival of subsets of glutamatergic neurons at posterior septal levels. The maturation status and integrity of remaining VG2^+ve neurons upon BSX loss remains unknown.

Studies using mice that lack BSX in the germline suggested a role for BSX in the hypothalamus and linked hypothalamic defects to locomotion deficits[52]. Our Bsx cKO mice allowed us to assess behaviours upon deletion of *Bsx* in septal neurons when hypothalamic expression of *Bsx* is intact. Before the initiation of behavioural testing, all mice (males and females) appeared normal with no visible impairments such as lacrimation, tremor, convulsions, piloerection, barbering, or other unusual behaviours. Bsx cKO mutant male and female mice had comparable weights to their littermate controls (Fig. 7a). Motor coordination and balance were assessed on a Rotarod apparatus. After one day of training, mice received three days of testing on accelerating rods. Both male and female Bsx cKO mice showed significant improvement throughout the three-day period, with no difference

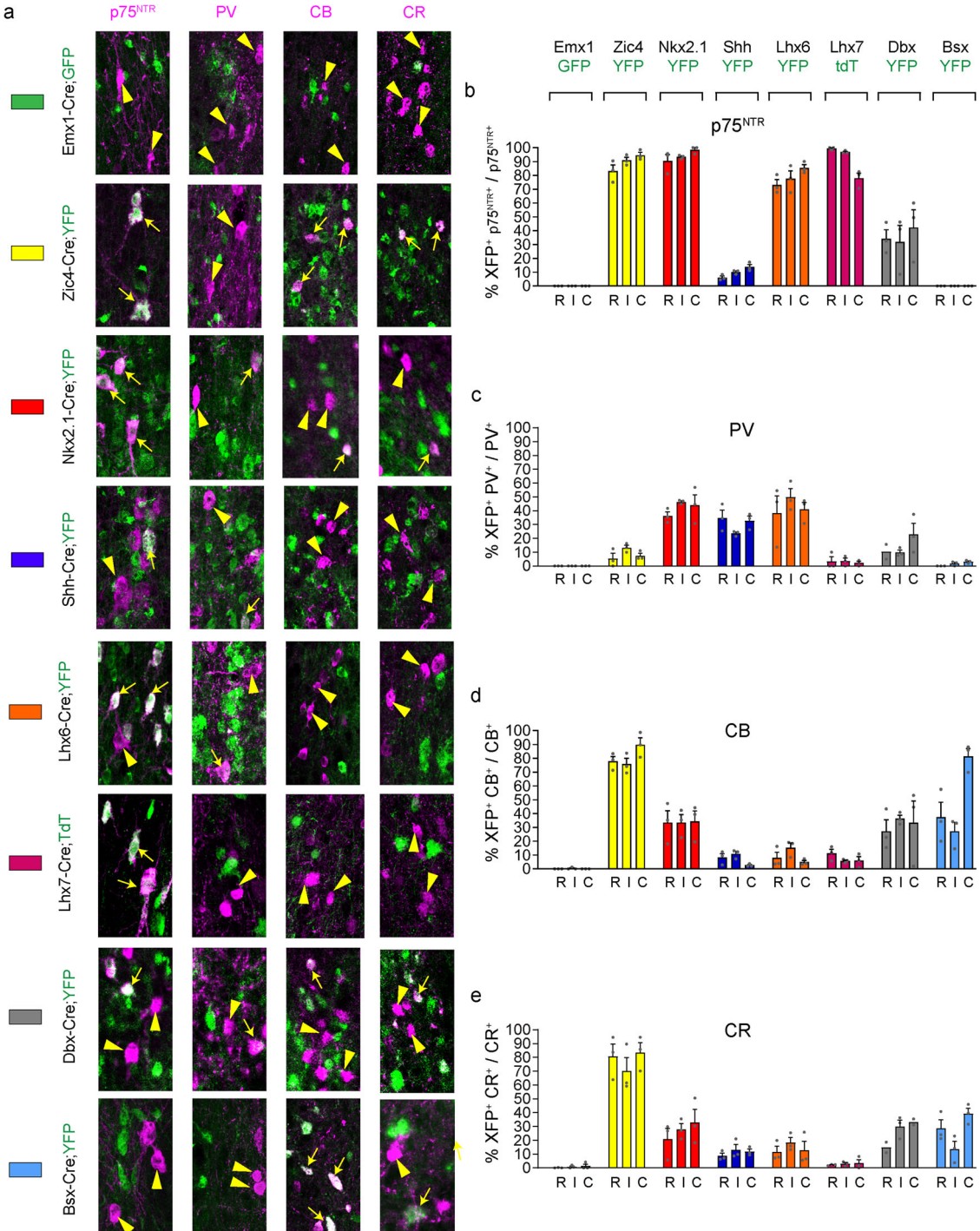

**Fig. 4 Septal and extra-septal embryonic origins of medial septal neurons. a** Contribution of different forebrain domains to MS populations. Double-labelling for p75NTR, CB, CR or PV and the fluorescent reporter protein GFP/YFP/TdT in the various transgenic lines at P30. White arrows and arrowheads indicate double and single labelled cells, respectively. **b–e** Histograms showing the percentage of neurons double labelled for GFP/YFP/TdT over the total population. Each transgenic line is identified by a different colour as shown in **a** and as summarised in Fig. 2d, e. $n = 3$ brains per transgenic line per marker except the following where $n = 2$ for Dbx-Cre;YFP mice: PV R, CR R and C. Histograms show Mean + SEM. Source data are provided in Supplementary Data 1. Scale bar: 25 μm.

in latency to fall compared to controls (Fig. 7b). To test general locomotion, animals were exposed to an open field task over 2–4 consecutive days (30 mins on day 1 and 10 mins on subsequent days). Compared to controls, mutant males travelled shorter distances on both session days 1 and 2 (Fig. 7c, d). However, there were no significant differences in average time spent moving or the average speed between control and mutant male or female

mice (Fig. 7e, f). We examined anxiety by calculating the time spent in the centre of the open field arena over the entire time spent moving (Fig. 7g). Male and female cKO mice spent a similar amount of time in the centre of the arena as their littermate controls (Fig. 7h–j), suggesting lack of anxiety defects. Although a statistically significant reduction in the time spent in the centre was observed in males on day 2, this did not prove to be

**Table 2 Fate-mapping summary.**

| | Emx1-Cre;GFP | | Zic4-Cre:YFP | | Nkx2.1-Cre;YFP | | Shh-Cre;YFP | |
|---|---|---|---|---|---|---|---|---|
| | LS (%) | MS (%) | LS (%) | MS (%) | LS (%) | MS (%) | LS (%) | MS (%) |
| p75 | – | 0 | – | 94 | – | 94 | – | 11 |
| PV | 0 | 0 | 73 | 8 | 40 | 42 | 0 | 30 |
| CB | 0.3 | 0.1 | 92 | 79 | 38 | 34 | 6 | 7 |
| CR | 1.2 | 0.8 | 94 | 78 | 8 | 27 | 3 | 11 |

| | Lhx6-Cre;YFP | | Lhx7-Cre;YFP/tdT | | Dbx-Cre;YFP | | Bsx-Cre;YFP | |
|---|---|---|---|---|---|---|---|---|
| | LS (%) | MS (%) | LS (%) | MS (%) | LS (%) | MS (%) | LS (%) | MS (%) |
| p75 | – | 79 | – | 91 | – | 36 | – | 0 |
| PV | 5 | 43 | 6 | 4 | 35 | 14 | 48 | 2 |
| CB | 8 | 9 | 19 | 8 | 22 | 32 | 7 | 49 |
| CR | 4 | 14 | 1 | 3 | 2 | 25 | 6 | 27 |

Summary of the septal contribution of different forebrain domains to LS and MS neuronal populations identified by expression of p75[NTR], PV, CB and CR. The different forebrain domains were labelled using the indicated Cre-expressing driver lines crossed to fluorescent reporters. Their spatial distribution is summarised in Fig. 2d, e.

significant over a 4-day period (Fig. 7j). Altogether the data suggest that deletion of *Bsx* in the septum does not lead to neurophysiological impairments, weight gain or anxiety, and does not affect coordination, balance or the ability to learn certain motor tasks. However, the reduced distance travelled in the open field test suggests reduced locomotion and voluntary movement.

To further explore the possibility that loss of *Bsx* in septal neurons affects voluntary locomotion, we chose a task that allows behavioural assessment in a more natural environment compared to the open field arena. The running wheel task is carried out in home cages containing a wheel and a detection system to automatically measure locomotion over 24 h during 8 consecutive days (see Methods section). Mice were single-housed in cages with running wheels in separate sessions for 8 consecutive days. All four groups, male and female control and mutants, significantly improved their daily wheel-running skills, reaching the maximal values on the 8th day (Fig. 7k–p). This indicates a normal ability to learn a certain motor task. However, both male and female mutants exhibited reduced performance, measured as cumulative distance travelled and average speed (Fig. 7k, m, n, p). Maximum speed was also significantly reduced for male Bsx cKO mice (Fig. 7o). These effects were more pronounced in male mice compared to females. Taken together, our behavioural analysis allows us to conclude that loss of BSX in septal glutamatergic neurons negatively impacts voluntary locomotion.

## Discussion

In the present study, we examined the embryonic origin and birthdate of septal neurons using newly-generated and existing genetic tools and distinguishing between LS and MSvDB populations. Most neurons originate from local septal precursors and are generated during early embryogenesis. An exception to this is the GABAergic MSvDB PV-expressing population which originates from surrounding extra-septal germinal zones. Glutamatergic septal neurons are also generated from local septal precursors and express the homeobox-encoding gene *Bsx* at embryonic stages. Loss of *Bsx* from septal glutamatergic neurons during embryogenesis causes selective loss of glutamatergic neuron subsets and deficits in voluntary locomotion.

Our data are consistent with previous birth-dating studies showing that the MSvDB has the earliest and shortest time of neurogenesis, while LS cells are generated at later time points over a prolonged time span[26,27,30]. This temporal bias results in a medio-lateral order of neuronal deposition and may lead to the emergence of the structural lamellar organisation of the adult MSvDB. Most septal neurons are generated after E10.5 and originate from resident precursors. PV MSvDB neurons are generated outside the septum at early embryonic stages (<E10 possibly) and reside in the most medial position of the MS. They are entirely distinct from PV LS neurons which have a septal origin and a glutamatergic phenotype. The external origin of MS PV neurons implies migration into the septum during embryogenesis. Such long-distance migration into the septum has also been identified for posterior septal glutamatergic neurons that populate the triangular septal nucleus and the bed nuclei of the anterior commissure. These neurons originate from diencephalic precursors in the thalamic eminence and migrate rostrally to enter the telencephalon[53]. GnrH-producing neurons also migrate through the septum from the nasal placode on their way towards the POA[54]. Thus, distinct birthdates, along with spatially distinct embryonic origins and long-distance migration, contribute to septal neuronal diversity.

Our study is limited in terms of lateral septal cell types examined, and this is largely due to technical limitations [neuropeptides such as somatostatin, enkephalin and oestrogen receptors previously shown to be expressed in the septum[19] have high turnover making their histological detection difficult, if not impossible, without a pre-treatment with toxic doses of colchicine], scarce availability of unique identifiers for different cells types, and lack of precise anatomical boundaries that define LS subnuclei. Transcriptomic analysis of septal neurons through single cell sequencing, combined with knowledge of anatomical projections and connectivity, will provide more refined molecular handles that will enable further dissection of the embryonic origin and function of LS neurons.

In addition to PV MSvDB neurons, the other prominent projections of the septo-hippocampal system originate from cholinergic and glutamatergic MS populations. We have previously explored the origins of the cholinergic MSvDB neurons[43]. The origin of glutamatergic neurons has not been examined. Their common neurotransmitter phenotype with cortical pyramidal neurons led to the suggestion that these neurons may have a pallial origin[55]. However, we demonstrate that these neurons have a partially overlapping lineage with transient glutamatergic migratory Cajal-Retzius cells[48] and originate from resident septal precursors.

Our findings indicate that, in contrast to the pallium and the subpallium, which generate neurons with district neurotransmitter phenotypes, the septum can generate glutamatergic, GABAergic and cholinergic neuronal cell types. It is unknown how this is orchestrated. It is possible that precursors are

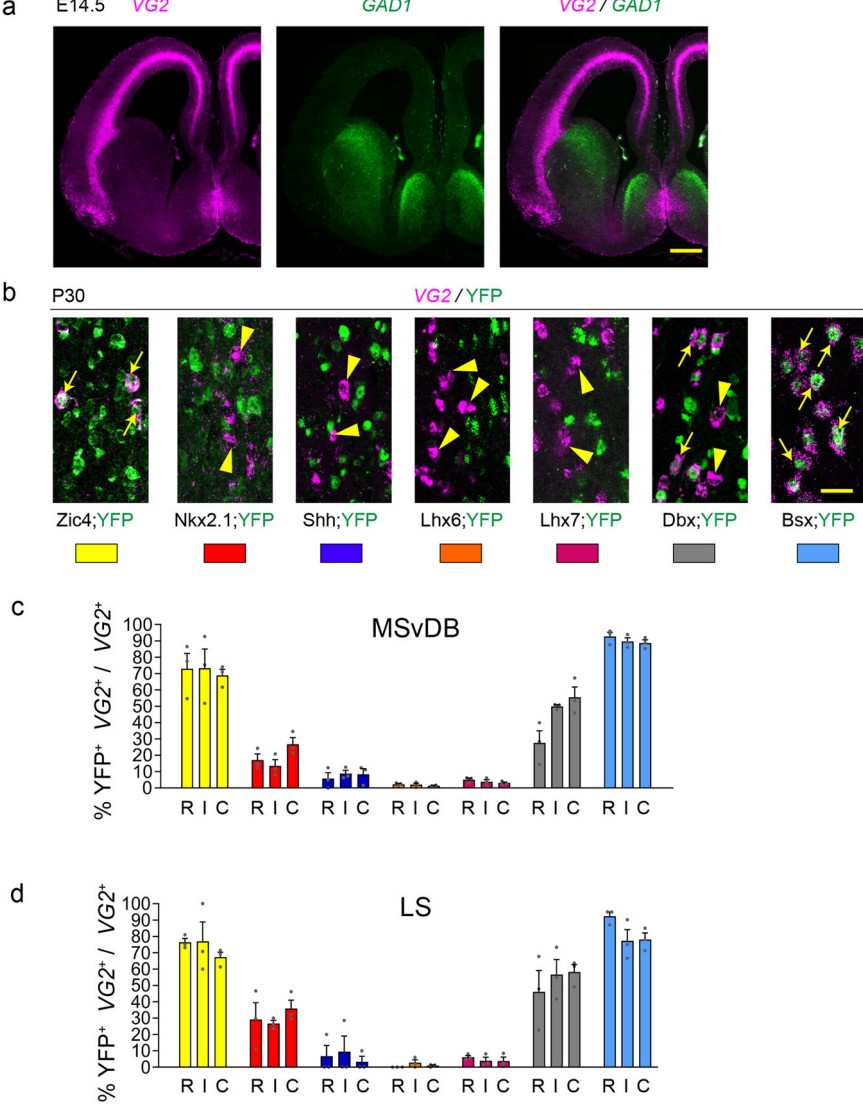

**Fig. 5 Septal origins of septal glutamatergic neurons with unique developmental codes. a** *VG2* and *GAD1* expression in the forebrain of a WT E14.5 mouse. **b** Contribution of different forebrain domains to *VG2* MS populations. Double-labelling for *VG2* and YFP in the various transgenic lines at P30. White arrows and arrowheads indicate double and single labelled cells, respectively. **c**, **d** Histograms showing the percentage of *VG2* neurons double labelled for YFP over the total *VG2* population in the MSvDB (**c**) and LS (**d**). Each transgenic line in **c**, **d** is identified by a different colour as shown in **b** and as summarised in Fig. 2d, e. $n = 3$ brains per transgenic line per marker. Histograms show mean + SEM. Source data are provided in Supplementary Data 1. Scale bars: **a** 200 μm; **b** 25 μm.

segregated within the septum. Alternatively, similar to cholinergic and GABAergic pallidal and striatal neurons, there may exist bipotential precursors whose differentiation is dependent on intrinsic factors[50,56]. A better molecular dissection of the embryonic septum will shed light on such questions of lineage and fate.

*Bsx* is a conserved homeobox gene expressed in different regions of the brain[49]. It has been implicated in mouse feeding and locomotion behaviour[52], mouse pup growth[57] and zebrafish pineal gland development[58,59]. *Bsx* does not regulate precursor patterning, but rather terminal differentiation and expression of genes such as *Npy* and *Agrp* in the hypothalamus[52,60] through direct binding to promoter regions[52,61]. Mice lacking BSX in the germline exhibit reduced locomotor activity and attenuated response to fasting, implicating BSX in locomotion and the control of energy balance[52]. In that study, locomotor deficits were attributed to hypothalamic defects, particularly in the NPY/AgRP neurons of the arcuate nucleus[52]. However, the possibility that

BSX-expressing neurons outside the hypothalamus may contribute to the phenotype had not been explored.

Our fate-mapping analysis identified BSX as a marker for septal glutamatergic neurons and our conditional deletion approach allowed us to evaluate its role in the septum and related behaviours in the absence of hypothalamic defects. The integrity of the hypothalamus in our septal Bsx cKO mice is supported by a number of observations: (1) Zic4-Cre and *Bsx* are expressed in largely non-overlapping regions of the developing hypothalamus, (2) *Npy* expression in the arcuate nucleus is unaltered in adult cKO and (3) body weight is unaffected in conditional mutant mice. By deleting septal *Bsx* while preserving hypothalamic expression, we find loss of septal glutamatergic neurons and associated reduced voluntary locomotion with preserved motor skills (see Rotarod). The loss of neurons and behavioural alterations are present in both male and female mice but are more pronounced in males. The reason for this sex-specific phenotype severity is unknown. Nevertheless, our

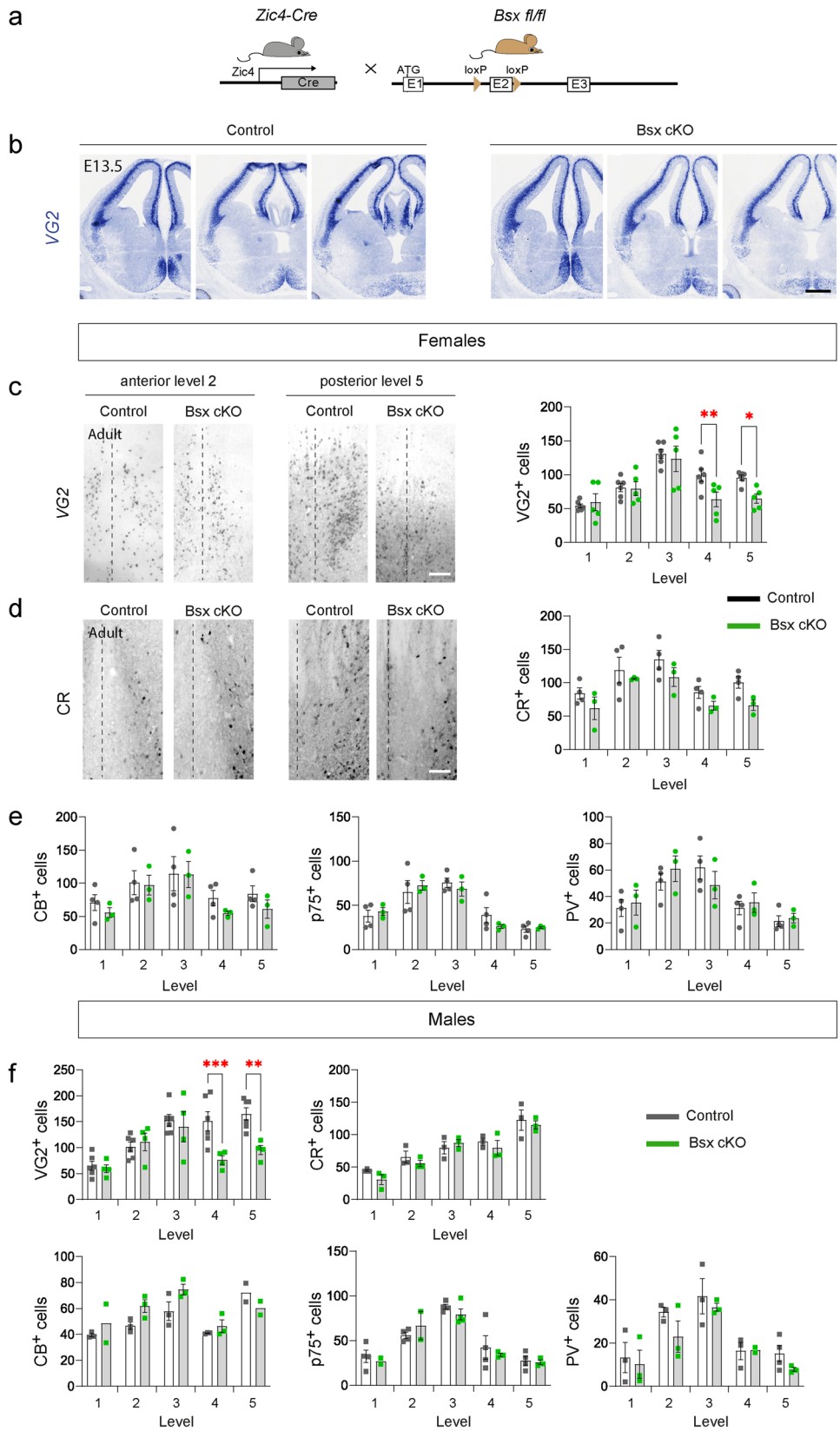

findings, together with the prominent role of glutamatergic MS neurons in voluntary movement and locomotion[24,62], allow us to propose that septal loss of BSX in glutamatergic neurons is sufficient to mediate locomotion deficits. Given the known interaction of the hypothalamus with the glutamatergic component of the MS[24,62,63], hypothalamic loss of BSX may contribute additional locomotion deficits in germline *Bsx* mutant mice. Finally, given the role of septal glutamatergic neurons in hippocampal theta oscillations[24], analysis of hippocampal network activity, and further behavioural testing would be needed to assess the impact of BSX loss on hippocampal dynamics and cognitive processing.

**Fig. 6 Essential role for BSX in medial septal glutamatergic neurons. a** Strategy for conditional deletion of *Bsx* in septal neurons. **b** Representative *in situ hybridisation* for *VG2* in control and Bsx cKO embryos at E13.5. **c** Representative expression of *VG2* at two anterior-posterior levels of the septum (levels 2 and 5) in female adult control and Bsx cKO mice and quantification of *VG2* neurons at five different anterior-posterior levels (anterior level 1 - posterior level 5). Control $n = 6$, Bsx cKO $n = 5$ mice at each level. 2-way RM ANOVA, Fisher's LSD test, level 4 $P = 0.009$, level 5 $P = 0.023$. **d** Representative expression of CR at two rostro-caudal levels of the septum (levels 2 and 5) in female adult control and Bsx cKO mice and quantification of CR neurons at five different anterior-posterior levels. Control $n = 4$, Bsx cKO $n = 3$. 2-way RM ANOVA, Fisher's LSD test. **e** Quantification of CB, p75 and PV neurons at five different rostro-caudal levels in female adult control and Bsx cKO mice. Control $n = 4$, Bsx cKO $n = 3$. 2-way RM ANOVA, Fisher's LSD test.
**f** Quantification of *VG2*, CR, CB, p75 and PV neurons at five different rostro-caudal levels in male adult control and Bsx cKO mice. *VG2*: $n = 6$ control, $n = 4$ Bsx cKO. CR, CB and PV, $n = 3$ mice of each genotype at each level except the following where $n = 2$: CB level 1 Bsx cKO, level 5 control and Bsx cKO and PV level 4 Bsx cKO. p75, control $n = 4$, Bsx cKO $n = 3$ except levels 1 and 2 Bsx cKO where $n = 2$. 2-way RM ANOVA, Fisher's LSD test. *VG2* level 4 $P = 0.0006$, level 5 $P = 0.0013$. All data show mean ± SEM. Source data are provided in Supplementary Data 1. Level 1: Bregma 1.18 mm; Level 2: Bregma 0.98 mm; Level 3: Bregma 0.74 mm; Level 4: Bregma 0.50 mm; Level 5: Bregma 0.26 mm. Scale bars: **b** 150 µm; **c, d** 100 µm.

The overlap between Zic4-YFP expression and *Bsx* in the epiphysis suggests that there is additional deletion of *Bsx* in the pineal gland in our Bsx cKO mice. BSX regulates the development of the pineal complex in zebrafish[60]. Therefore, it is possible that the effect in locomotion may be partially mediated by alterations in circadian rhythms. BSX is also required for the expression of the corticotropin-releasing hormone ligand gene, uts1, in the zebrafish septum[60]. Whether the orthologue of this neuromodulator gene (*Ucn*) is also expressed in the mammalian medial septum and maps to glutamatergic neurons needs to be determined. As the function of BSX in mammalian septal neurons is entirely unexplored, further experiments are required to characterise the molecular pathways regulated by this transcription factor and which contribute to the regulation of locomotion.

In conclusion, our findings demonstrate a predominant septal origin of septal neurons with a substantial contribution from neighbouring precursor regions. Successive neuronal temporal specification together with migration of neuron subsets into the septum contribute to the generation of septal neuronal diversity. The transcription factor BSX is an early marker for septal glutamatergic neurons and is essential for their development/maturation and contribution to locomotion. Further dissection of septal neuronal diversity and function is needed to understand the multiple roles of septal neurons and their contribution to diverse behaviours.

## Methods

**Transgenic mice.** Emx1-Cre (MGI:3761167)[42], Zic4-Cre (MGI:4840322)[43], Nkx2.1-Cre (MGI:3761164)[42], Shh-Cre (JAX 005622)[64], Lhx6-Cre (JAX 026555)[45], Dbx1-Cre (MGI:3757955)[48], Nestin-Cre (JAX 003771) and three reporter mice, Rosa26R-GFP (JAX 004077)[65], R26R-YFP (JAX 006148)[66] and Rosa26R-tdTomato (JAX 007914)[67] used in this study have been described previously. Crosses between Cre-expressing mice (e.g. Zic4-Cre) and fluorescent reporters (e.g. Rosa26R-YFP) are abbreviated in the manuscript (e.g. Zic4-Cre;YFP or Zic4;YFP). Mice expressing Cre under control of the *Lhx7* or *Bsx* were generated using bacterial artificial chromosome (BAC) transgenic technology[68]. Details on the generation of the previously reported Lhx6-Cre mice are also provided in this study. The codon-improved Cre recombinase (iCre)[69] containing a nuclear localisation signal was fused to the initiation codon of each of the genes using a PCR-based approach. This was followed by a Simian Virus 40 polyadenylation signal and a Kanamycin resistance cassette that was flanked by FRT sites for selection of recombinant BACs. BAC recombination was designed to delete the remaining ATG-encoding exon together with 50-200 bp from the downstream intron. The following BACs were used: Lhx7-Cre: 214 Kb BAC RP23-379E17 containing 130 Kb upstream and 61 Kb downstream of the Lhx7 gene; Lhx6-Cre: 235 Kb BAC RP24-384G1 containing 137 Kb upstream and 80 Kb downstream; Bsx-Cre: 180 Kb BAC RP24-255N16 containing 70 Kb upstream and 110 Kb downstream. Kanamycin resistance was removed in vitro, and the BACs were linearised and purified prior to microinjection into fertilised eggs. *Bsx* conditional knock-out mice were obtained as Knock-out first, promoter driven (tm1a) frozen embryos from UCDavies KOMP Repository. They can now be obtained from MMRRC (Bsx$^{tm1a(KOMP)Wtsi}$ MMRRC:052869-UCD, MGI:4363233). 'Knock-out first' mice were converted to conditional deletion mutants (tm1c) using germline recombination with a FLP-expressing mouse. This results in *loxP* sites inserted in intron 1 and intron 2. Loss of exon 2 through Cre excision (and generation of the tm1d allele) was confirmed using a PCR-based approach. Primer sequences used for iCre and *Bsx* allele detection are as follows: iCreF (iCre250S): 5′ GAG GGA CTA CCT CCT GTA CC 3′; iCreR (iCre 880AS): 5′ TGC CCA GAG TCA TCC TTG GC 3′; Int1F1 (BsxInt1F1): 5′

CAA CCC TGC TAC TGA CAA GG 3′; Int1R1 (BsxInt1R1): 5′ CTT CCA GTT ATC TGT TAG GCC 3′; Int2R1 (BsxInt2R1): 5′ GGT TCT GGG CCA GCC CTG GGC 3′. Loss of function in tm1a mice was confirmed by loss of hypothalamic NPY expression in the 'knock-out first' mice[52]. All mice used in this study were maintained on a C57BL/6J/CBA background. Mouse colonies were maintained at the Wolfson Institute for Biomedical Research, University College London, following UCL ethical approval and in accordance with United Kingdom legislation (ASPA 1986).

**Tissue preparation.** The day of the vaginal plug was considered E0.5, and the day of birth was considered day 0. Whole embryo heads (for embryos E13.5 and younger) were fixed overnight in 4% (w/v) paraformaldehyde (PFA) in PBS. Postnatal animals were anesthetized and perfused first with saline (0.9 % NaCl) followed by 4% (w/v) PFA through the left ventricle of the heart. Adult brains were dissected out, sliced into 2 or 3 mm-slices using a mouse brain coronal matrix (PlasticsOne), and post-fixed in 4% PFA overnight. Fixed samples were cryoprotected overnight by immersion in 20% (w/v) sucrose in PBS. All samples were embedded in Tissue-Tek OCT compound (R. A. Lamb Medical Supplies, Eastbourne, UK), frozen on dry ice, and stored at −80 °C.

**Immunohistochemistry.** Embryonic brains were cut on a cryostat into 20-µm-thick coronal or horizontal sections (E13.5) and collected directly on Superfrost plus slides (BDH Laboratory Supplies, Poole, UK). Adult animals were anaesthetized and transcardially perfused with 4% PFA in PBS, the brains were extracted and post-fixed overnight in 4% PFA at 4 degrees, and then transferred in PBS. Adult sections were cut coronally (30 µm thickness) and were serially collected in PBS for free floating procedure. All sections were blocked in PBS containing 10% heat-inactivated sheep serum (Sigma, St. Louis, MO) and 0.1% Triton X-100 (Sigma) at room temperature for 1h. Immunohistochemistry was performed with the following primary antibodies: rat anti-GFP IgG2a (1:1000; Nacalai Tesque, Kyoto,Japan, #04404-84); mouse anti-calbindin (#300), rabbit anti-calbindin (#CB38a), rabbit anti-calretinin (#7697), mouse anti-calretinin (#6B3), (all 1:1000 from Swant, Bellizona, Switzerland); mouse anti-parvalbumin (1:1000, Chemicon Millipore #MAB1572), rabbit anti-p75$^{NTR}$ (1:1000; Promega, Southampton, UK, #G3231), rabbit anti-TTF-1 (NKX2-1) (1:100 Santa Cruz Biotechnology, CA, #sc-13040) and rabbit anti-Gsx2 (1:500 Millipore #ABN162). Primary antibodies were applied overnight at 4 °C. Secondary antibodies used were AlexaFluor 488- conjugated, AlexaFluor 568-conjugated, and AlexaFluor 647-conjugated donkey anti-rabbit IgG or donkey anti-rat IgG or donkey anti-mouse IgG (all used at 1:1000; Invitrogen, Carlsbad, CA) and were applied for 60 min at room temperature together with Hoescht 33258 (1:10 4; Sigma) to detect cell nuclei. All secondary antibodies were diluted in block solution (1:1000). Floating sections were transferred onto Superfrost plus slides (BDH Laboratory Supplies) and air dried. All sections were coverslipped with Dako fluorescent mounting medium. For detection of markers on Bsx cKO and control sections, endogenous peroxidase activity was quenched with 0.6% $H_2O_2$ for 20 minutes and antibodies were applied overnight as described above. A biotin-conjugated secondary antibody (donkey anti-rabbit IgG, 1:500; Millipore) was used to detect the primary antibodies followed by the Avidin/Biotinylated enzyme Complex (ABC) reaction (Vectastain ABC kit, Vector Laboratories) prepared according to manufacturer's instructions, and addition of DAB reagent (Vector Laboratories). After DAB reaction, all slides were dehydrated in progressively increasing concentration of ethanol and xylene, and were mounted with DPX mounting media.

**RNA In situ hybridization (ISH).** Fixed samples were cryoprotected for >12 hours by immersion in 20%(w/v) sucrose/PBS pre-treated with diethyl pyrocarbonate (DEPC) (Sigma), embedded in OCT (Tissue Tek; Raymond Lamb Ltd Medical Supplies) and frozen on dry ice by immersion in isopentane. All samples were stored at −80 °C until needed. Sections were collected on Superfrost Plus slides (VWR International) and allowed to air dry before hybridising overnight at 65 °C (in a chamber humidified with 50% v/v deionized formamide and containing 1x SSC buffer) in a buffer containing digoxigenin (DIG)- labelled antisense RNA probe (prepared according to manufacturer's instructions and diluted 1:1000 in hybridisation buffer). Hybridisation buffer consisted of 50% v/v deionized

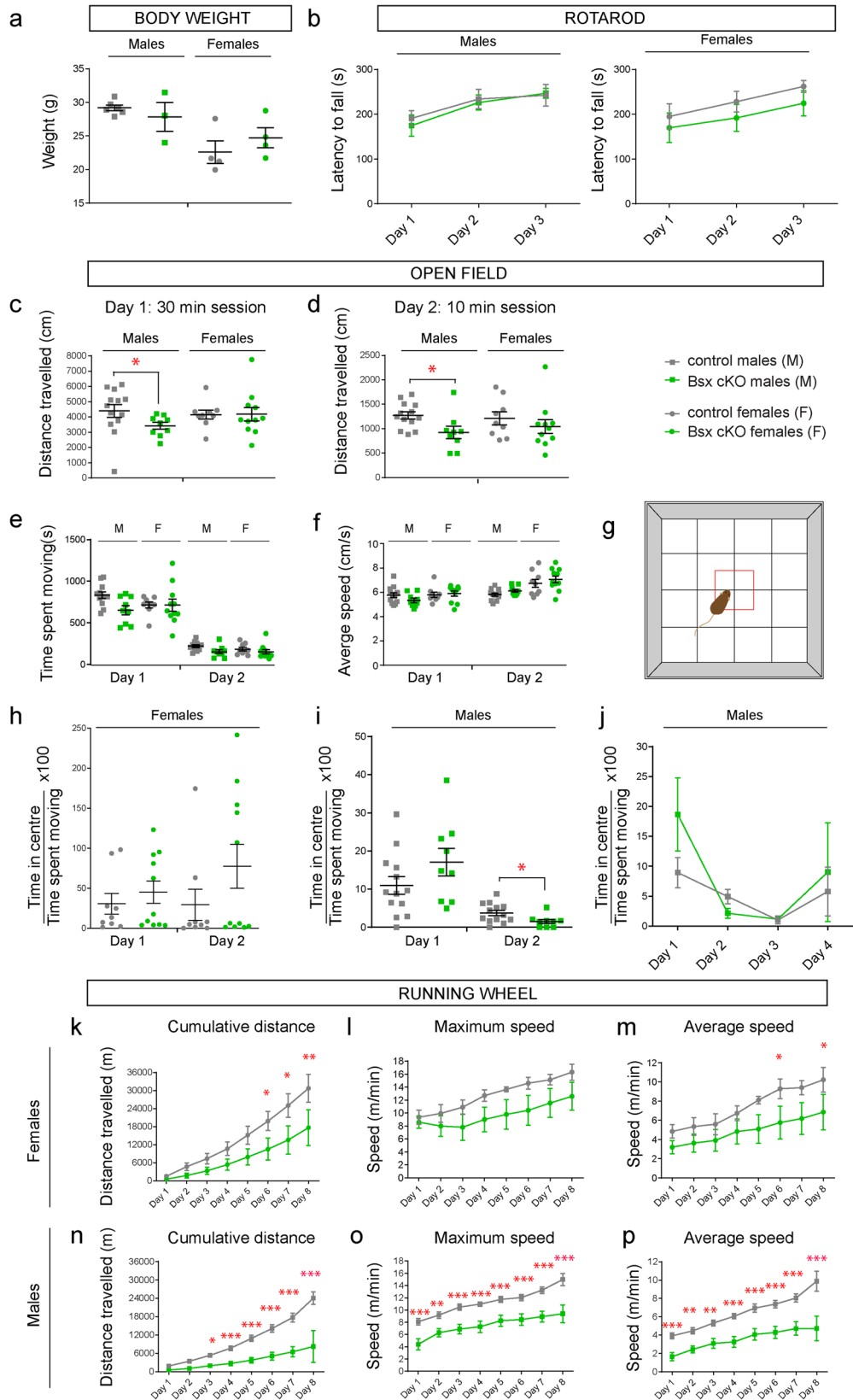

formamide (Sigma), 1x "Salts" [2 M NaCl, 50 mM EDTA, 100 mM Tris-HCl, pH 7.5, 50 mM NaH$_2$PO$_4$2H$_2$O, 50mM Na$_2$HPO$_4$, 0.1 mg/ml tRNA from baker's yeast (phenol chloroform extracted, Roche Diagnostics)], 1x Denhardt's solution (Sigma), and 10% w/v dextran sulphate (pre-dissolved in DEPC-treated water and maintained at 4 ℃ as 50% stock). Sections were washed three times at 65 ℃ for 30 min each in pre-warmed wash solution (50% v/v formamide, 1x SSC, 0.1% Tween 20), followed by two washes in 1x MABT (100 mM maleic acid, 150 mM

NaCl, pH 7.5, 0.1% Tween 20) at room temperature also for 30 min each. Blocking was performed using 2% w/v Blocking Reagent (Roche Diagnostics), 10% v/v heat-inactivated sheep serum (Sigma) in 1x MABT for 1 hour at room temperature and anti-DIG antibody conjugated with alkaline phosphatase (AP) (Roche Diagnostics) diluted 1:1500 in blocking solution was applied overnight at 4 ℃. Following washes in 1x MABT, sections were equilibrated in 100 mM NaCl, 100 mM Tris-HCl, pH 9.5, 0.1% Tween 20 for 10 min. Development was performed at 37 ℃ for 4–8 h

**Fig. 7 Behavioural assessment of septal Bsx cKO mice: essential role for *Bsx* in locomotion. a** Body weight: no differences in body weight values found between control and mutant mice. Males: controls $n = 6$, Bsx cKO $n = 3$. Females: controls $n = 4$, Bsx cKO $n = 4$. Unpaired *t*-test with Welch's correction. $P > 0.05$. **b** Rotarod: latency to fall from the Rotarod throughout a 3-day test is shown. Males: controls $n = 8$, Bsx cKO $n = 6$. Females: controls $n = 7$, Bsx cKO $n = 6$. 2-way RM ANOVA. No significant effect of genotype. Gradual improvement was detected for all mice indicating that motor learning has occurred (Effect of Day, 2-way RM ANOVA, Males, $P = 0.0003$; Females, $P < 0.0001$). **c–j** Open field task. **c–d** distance travelled over a 2-day test. Only mutant males show reduced distance travelled on both days compared to controls. Mann Whitney $U$ test, Day 1, $P = 0.025$; Day 2, $P = 0.014$). **e, f** similar time spent moving and average speed between control and mutant mice over a 2-day test. **g** diagram showing the open field arena. The centre of the arena is shown as a red box. **h** similar time spent in centre between female control and mutant mice. **i**, male mutants showed reduced time in centre on day 2 (Mann Whitney U test, Day 2, $P = 0.0215$) but this was not significant over a 4-day test period (**j**). **c–i** Males: controls $n = 13$, Bsx cKO $n = 9$. Females: controls $n = 9$, Bsx cKO $n = 11$. j: controls $n = 6$, Bsx cKO $n = 5$. **k–p** Running wheel task: female and male BSX cKO mice show reduced cumulative distance run on the wheel. Females: $n = 4$ mice for each genotype. 2-way RM ANOVA, Fisher's LSD test $*P < 0.05$, $**P < 0.01$, $***P < 0.001$. Distance: effect of Day $P < 0.0001$. Max speed: effect of Day $P < 0.0001$. Average speed: effect of Day $P < 0.0001$. Males: controls $n = 11$; Bsx cKO $n = 9$ except day 8 where $n = 3$ for both groups. Distance: effect of Genotype $P < 0.0001$, Day $P < 0.0001$ and Interaction $P < 0.0001$. Max speed: effect of Genotype $P = 0.0004$ and Day $P < 0.0001$. Average speed: effect of Genotype $P = 0.0002$ and Day $P < 0.0001$. All graphs show mean ± SEM. Source data are provided in Supplementary Data 1.

with nitroblue tetrazolium/5-bromo-4-chloro-3-indolyl phosphate in freshly prepared staining solution containing 100 mM NaCl, 50 mM MgCl2, 100 mM Tris-HCl, pH 9.5, and 0.1% Tween 20. To increase sensitivity, 5% (w/v) polyvinyl alcohol was included during staining. For fluorescent detection of ISH signal, DIG-labelled (or FITC-labelled) RNA probes were detected with an anti-DIG (or anti-FITC) POD-conjugated antibody followed by incubation with Tyramide-Cy3 (or Tyramide Cy5) (1:100, TSA-Plus, PerkinElmer) for 10 min – 3 hours at room temperature. FISH was sometimes followed by IHC as follows: FISH detection was stopped with 3% $H_2O_2$ before sections were used for immunodetection as described in the previous section. The following plasmids were used to generate RNA probes: IMAGE clone 374236 for *Gad1* (*Gad67*) (linearised with *Xho*I and transcribed with T3), a 0.7 kb DNA fragment corresponding to the 3' UTR sequence of the mouse *Bsx* gene and cloned into pCRIITopo (linearised with *Bam*HI and transcribed with T7), a 2.5 kb fragment from mouse *vGlut2* (*Slc17a6*) cDNA cloned into pT7T3-PacI (a kind gift from T.Jessell, Columbia University, USA) (linearised with *Eco*RI and transcribed with T3) and a 1 kb cDNA for iCre cloned into pBluescriptII (linearised with *Eco*RV and transcribed with T7).

**EdU birthdating.** 5-ethynyl-2′-deoxyuridine (EdU, Molecular Probes) was dissolved in sterile PBS at 2 mg/ml. Pregnant females were administered five intraperitoneal injections of EdU (10mg/Kg body weight) at two-hour intervals starting at 10:00 am. The pups were perfused at P30 and the tissue processed as described above with the exception of tissue fixation that was performed for 45 min at room temperature in 4% PFA. EdU detection was carried out after immunohistochemistry for the various markers using the Click-iT EdU Alexa Fluor 647 Imaging Kit (Molecular Probes) according to manufacturer's instructions.

**Image processing.** Images were captured with a Zeiss fluorescent microscope or with a Leica confocal microscope. Images were further processed with Adobe Photoshop (Adobe Systems Inc., San Jose, CA) for general contrast and brightness enhancements. The final compositing of the figures was performed with Adobe Illustrator (Adobe Systems Inc., San Jose, CA). Images of RNA ISH were taken using a ZEISS Axio Scan.Z1 and processed using ZEISS ZEN lite software.

**Mouse behaviour.** Two cohorts of male and female mice (from the age of 3 months) have been assessed on a battery of behavioural tests. All experiments took place between 09:00 to 17:00 in a room where external sounds were masked by white noise. The mice were left in the room for 30 minutes before each test session in order to minimise distress caused by the transportation. The tests were performed in the following sequence: neurophysiological assessment, open field, rotarod, running wheel.

An initial neurophysiological screening consisting of very short tests to assess broadly sensory and motor function, and general health was performed on all animals two weeks before the start of the tests.

*Open field.* Mice were exposed to open field (OF) over two days. Day 1: 30-min session in the non-transparent arena (dimensions: $30 \times 30 \times 40$ cm); Day 2: animals were exposed to the open field for 10 min. For a subgroup of male mice, the OF task was extended for 2 extra days with one 10 min-sessions each day. Four main parameters: distance travelled, time spent moving (s), mean speed (cm/s), time spent in central area (s) were obtained for further analysis. Tracking of the mice was carried out with ActualTrack software (Actual Analytics, Edinburgh, UK).

*Rotarod test.* Evaluation of fine motor coordination and balance was assessed via Rotarod apparatus over four days. On the first day, mice underwent one training session consisting of 3 trials of 120 seconds each. Rotating rod spinning speed was kept constant

(4 revolutions per minute). During the subsequent 3 test days the rotating rod was set to accelerate from 4 to 40 revolutions per minute over 300 s. Mice were given 15 min of rest between the trials for them to fully recover. Latency to fall was registered once the mouse landed on a lever. No differences in weight were observed between the groups indicating that this factor did not have an effect on the task.

*Running wheels.* As a further measure of motor performance, mice were selectively assigned to a running wheel test according to their weight and Rotarod performance. For this, animals were exposed to the housing cages with a voluntary access to the attached running wheels and 24 hours access to food and water. Running wheels were equipped with irregularly spaced crossbars[70]. One wheel revolution corresponds to a running distance of 0.38 m. A rotation sensor was connected to the wheel axis and the wheels were connected to the automated recording systems that allowed calculating average speed, maximum speed and total distance run. Locomotion of the animals was assessed throughout 8-day period. Recordings were collected at a 1-min-interval between 6 p.m. and 7 a.m. (dark cycle), and 1-h-interval between 7 a.m. and 6 p.m. (light cycle).

**Statistics and reproducibility.** Experiments presented in this study were repeated independently a minimum of two times. For all quantification experiments a minimum of three mice was used for each genotype. For all experiments animals are considered as biological replicates and for cell counts, sections and levels are technical replicates. Precise *n* numbers used in each experiment is specified in the figure legends. The extent of co-localisation between GFP/YFP/tdTomato and the various markers in each of the transgenic mice was determined as follows. Counts were performed at three different rostro-caudal levels corresponding approximately to Bregma 1.18 (rostral section), Bregma 0.74 (intermediate section) and Bregma 0.26 (caudal section) for each brain. On each section the midline was drawn, and counts were performed on both sides separately for each nucleus (MSvDB and LS) and averaged per slice. Absolute numbers of cells counted per level is shown in Table 1. For comparison of *Bsx* cKO and controls, counts were performed at five different rostro-caudal levels corresponding approximately to Bregma 1.18 mm Level 1; Bregma 0.98 mm Level 2; Bregma 0.74 mm Level 3; Bregma 0.50 mm Level 4; Bregma 0.26 mm, Level 5. Where possible, cell counts and behavioural experiments were performed by investigators blind to the genotype. Statistical analysis was performed in GraphPad Prism (GraphPad Software, La Jolla, CA, USA). All data were tested for normality using a Kolmogorov-Smirnov or Shapiro-Wilk tests and subsequently analysed using an appropriate statistical test: Unpaired *t*-test with Welch's correction or two-way ANOVA with *post hoc* uncorrected Fisher's Least Significant Difference (LSD) test for normally distributed data; and the nonparametric two-tailed Mann–Whitney U test for non-normally distributed data.

**Reporting summary.** Further information on research design is available in the Nature Research Reporting Summary linked to this article.

## Data availability

All data generated or analysed during this study are included in this published article and its supplementary information files. Source data for all the figures are provided in Supplementary Data 1.

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

## Acknowledgements

We thank our colleagues at the Wolfson Institute for Biomedical Research (University College London) for helpful comments and discussions. We thank A. Pierani for providing mouse Dbx-Cre tissue for fate-mapping. A.N.R. was supported by a PhD studentship from the Wellcome Trust. Financial support for the work was provided by the European Research Council (Grant agreement 207807), the UK Biotechnology and Biological Sciences Research Council (BB/N009061/1) and the UK Wellcome Trust (108726/Z/15/Z).

## Author contributions

Conceptualisation by L.M. and N.K.; Investigation by L.M., Z.A., M.A., Y.M., N.B-V., A.N.R. and N.K. Writing by L.M. and N.K. Funding acquisition by N.K.

## Competing interests

The authors declare no competing interests.
