## [Peer Review File · Communications Biology]

Reviewers' comments:

Reviewer #1 (Remarks to the Author):

The manuscript by Kessar and colleagues explores the origin of the different types of neurons present in the forebrain septum of the mouse brain. They proficiently used a combination of immunofluorescence assays, EdU birth-dating, genetic fate mapping, and loss-of-function approaches (notably, some of mouse line are newly generated) to demonstrate the developmental code of generation of those neurons. The work is well executed and written and can be a resource for those interested in the analysis of this brain area. I am convinced that, after some adjustments, will be suitable for publication in *Communication Biology*. Here below, some points to address for: (i) helping to follow the experimental flow, (ii) circumstantiating and better disclosing key data, and (iii) enlarging the discussion.

- Figure 1a-g: quantification of the overlap seems needed here
- Figure S2: please indicate the age of the embryos in the figure
- Figure S2a: please highlight the septum for non-expert readers
- The authors said that "Cholinergic and GABAergic are entirely distinct" however Lhx7-Cre (known to be a cholinergic marker) labels a considerable number of CB neurons in LS (Fig 2). Can the author discuss this discrepancy?
- Figure 5d: tendency is not difference, so the question here is: Is this numerosity sufficient to grab differences? if yes and there is no difference, then no claim on CR neurons (and I would remove the sentence).
- Is there any sign of neurodegeneration in the septum of BSX KO animals or is a problem of no/miss-specification. May the authors include a quantification of cell death within the septum of their model?
- Behavior: the authors said that exploratory behavior is not affected in BSX septal KO, however the short distance that mutants travelled may indicate a less pronounced tendency to explore the cage. I would recommend other tests to assess exploration and anxiety levels (e.g. elevated plus maze)
- May the authors discuss better the fact that the PV neurons are so different between LS and MSvDB (excitatory vs inhibitory neurons)?

Reviewer #2 (Remarks to the Author):

In their manuscript, Magno et al., investigate the origin of forebrain septal neurons and the role of the Bsx homeodomain transcription factor in septal glutamatergic neurons. The authors first characterise the huge diversity of glutamatergic, GABAergic and cholinergic neurons, their localisation within the septum and their birthdates. To identify the origin of these diverse sets of neurons, the authors employed a Cre transgene based fate mapping strategy. Using a number of novel and previously established Cre driver lines, they found that most septal neurons originate from distinct progenitor domains in the septum, while a population of GABAergic, Parvalbumin expressing neurons are generated outside of the septum. Finally, they investigate the role of the BSX transcription factor which is expressed in almost all septal glutamatergic neurons through conditional inactivation. These BSX conditional mutants present with a spatially limited reduction in glutamatergic neurons and with locomotion defects.

Taken together, the authors use state of the art techniques to address important questions in the development of the septum and the hitherto uncharacterized role of a specific septal subpopulation of neurons in septum controlled behaviours. Their findings are supported by detailed and very carefully performed analyses in their well written manuscript. Therefore, I recommend publication in Communications Biology, however, the authors need to address a couple of points before acceptance.

1) The authors convincingly demonstrate a locomotion phenotype in Bsx conditional mouse mutants but no data are presented that Bsx was indeed deleted from the septum and that the deletion is confined to Zic4 lineage cells. The authors should perform immunofluorescence analysis of Bsx in control and mutant brains to demonstrate in which brain regions Bsx protein expression is lost. Moreover, to make the author's conclusion stronger that glutamatergic neurons in the septum play a crucial role in locomotion, a better comparison of the locomotion phenotypes in global and conditional Bsx mutants would be very informative.

2) There are some discrepancies between the co-immunostainings and the quantification presented in Figures 2, 3 and 4. For example, according to the quantification in Figure 3, a certain degree of CR+ neurons originate from the BsxCre lineage, but the figure shows no overlap. The authors should carefully revise the immunostainings in these three figures to correspond with the quantifications.

Reviewer #3 (Remarks to the Author):

In this manuscript, Magno et al. present a detailed immunohistochemical overview of the various lamellae of the septal complex and then proceed to identify the embryonic origins for each of the subpopulations characterized. In the course of their experiments, the authors utilize 2 newly generated mouse driver lines, Lhx7-Cre and Bsx-Cre and a new conditional allele, BSX cKO. Finally, the authors describe a novel role for Bsx, whose absence results in a significant reduction in glutamatergic neurons in the posterior MSvDB.

Overall, the work is very carefully and rigorously performed. The proper genetic controls are used throughout. Its significance to septal biologists and telencephalic neurobiologists is high. However, there are a number of impediments that could be improved upon with a re-write. The most prominent are the description of the neuroanatomy and the Cre driver lines used to interrogate its origins. Here, the narrative flow of the manuscript was not easy to follow and I offer the following suggestions for how to improve clarity:

- 1) A basic schematic describing the connections from the septal complex to other parts of the brain
- 2) An explanation of why the immunomarkers used were chosen. What populations do they label?
- 3) Relatedly, clarify upfront that P75NTR neurons are cholinergic
- 4) Incorporate Supplemental Figure 4C into main figures, either as a summary of the results at the end or at the beginning to describe what the Cre driver lines label, or both if you prefer.
- 5) Keep a consistent format for how the driver lines fate map each immunomarker. Figure 2a and b was probably the clearest format (as opposed to 3a-e or 4b-d).

The other major issue was to do with behavioral analysis of Zic2-Cre; Bsx cKO. It appears that only male mutants exhibit a behavioral phenotype. This was not commented upon either in the results or the discussion. A question related to this: if one were to deconvolve the histological data from Figure 5c by sex, is there a more pronounced effect on cell loss with male mutants? Regardless, this finding needs to be addressed directly in the manuscript.

Minor Points:

- The dot plot onto the septal diagram in Fig 1g: Is this one representative slice? Would it be more

informative to compile all of the cell placements for all experiments to get a truer sense of cell type placement within lamellae?

- Figure 1h: indicate the boundary between the MSvDB and the LS

- Page 4, middle it reads: "The laminar organization of the MS is rearranged at the ventral tier of the vDB". It is unclear what rearranged pertains to.

- Page 7, second paragraph it reads: "BSX is an evolutionary conserved homeobox-encoding gene..." It should be evolutionarily.

- Figure 5c, d: Clarify in the Results text and in the Figure Legend that levels 1-5 indicate anterior to posterior.

REVIEWER COMMENTS and *OUR RESPONSES*

We thank the reviewers for their enthusiasm, their thorough assessment of the manuscript and their insightful and helpful comments. We addressed all concerns either experimentally or through discussion and clarification. We hope the reviewers will find the revised manuscript significantly improved. A point-by-point response follows:

Reviewer #1:

1. "Figure 1a-g: quantification of the overlap seems needed here"

We present the quantifications as illustrations in a revised Figure 1h. Briefly: In the MS at the three levels examined (R, I and C), PV and p75^{NTR} are entirely distinct populations and occupy different positions within the MS. PV and CB are also distinct and again occupy different positions within the MS. There are only small overlaps between PV and CR and between p75^{NTR} and CB. CB and CR also partially overlap. Quantification of the overlaps is described in the revised manuscript. In the Lateral septum PV does not overlap with CB or CR. A small overlap exists between CB and CR.

2. "Figure S2: please indicate the age of the embryos in the figure"

This has been done (this is now Figure 2 in the revised manuscript).

3. "Figure S2a: please highlight the septum for non-expert readers"

This has been done (this is now Figure 2 in the revised manuscript).

4. "The authors said that "Cholinergic and GABAergic are entirely distinct" however Lhx7-Cre (known to be a cholinergic marker) labels a considerable number of CB neurons in LS (Fig 2). Can the author discuss this discrepancy?"

Indeed, there are no p75^{NTR+ve} cholinergic cells (also confirmed by ChAT staining) in the LS and yet Lhx7-Cre labels some CB cells. Lhx7 is involved in the determination of cholinergic versus GABAergic cell fate (doi:10.1242/dev.038083). Since Lhx7-Cre shows the history of Cre expression, it may label some non-cholinergic cells that may have expressed low levels of the Cre transgene but never became cholinergic. This is now clarified in the manuscript in the second paragraph of the section 'Septal and extra-septal neuroepithelial zones generate LS and MSvDB neurons' lines 166-171.

5. "Figure 5d: tendency is not difference, so the question here is: Is this numerosity sufficient to grab differences? if yes and there is no difference, then no claim on CR neurons (and I would remove the sentence)."

The number of mice used is sufficient to detect significant changes. We removed the sentence and made no claim for CR neurons. The lack of significant difference between control and cKO mice in the number of CR cells is observed in both male and female mice. This is shown in Figure 6d, f.

6. "Is there any sign of neurodegeneration in the septum of BSX KO animals or is a problem of no/mis-specification. May the authors include a quantification of cell death within the septum of their model?"

While this would be interesting to check, we do not have the capacity to do it. We have not attempted to determine what happens to septal neurons in the absence of Bsx. That would be something to do in a follow-up study. There may indeed be some re-specification and/or cell death, as the reviewer points out. In either case, given the postmitotic expression of Bsx in these neurons, we anticipate a protracted period of neuronal death in postnatal mice as incorrectly specified neurons may fail to integrate into networks and, as a consequence of lack of survival signals (such as activity) they may eventually die. Such slow neuronal death will be very difficult to detect using standard methods given the rapid clearance of dead cells.

7. "Behavior: the authors said that exploratory behavior is not affected in BSX septal KO, however the short distance that mutants travelled may indicate a less pronounced tendency to explore the cage. I would recommend other tests to assess exploration and anxiety levels (e.g. elevated plus maze)"

We quantified other parameters in the open field arena in order to assess anxiety. This is shown in figure 7g-j. Both males and females spent a normal amount of time exploring the centre of the arena, indicating lack of anxiety. Additionally, we tested male mice for 2 extra days in the open field arena and observed no difference between mutants and controls (Figure 7j). Nevertheless, we removed the statement that 'exploratory behaviour was not affected' because we do not have the capacity to perform other tests such as the elevated plus maze suggested by the reviewer. In addition, it is not clear to us that behaviour in the elevated plus maze would not be affected by a deficiency in locomotion.

8. "May the authors discuss better the fact that the PV neurons are so different between LS and MSvDB (excitatory vs inhibitory neurons)?"

This is now mentioned in the results, paragraph 5 (lines 131-135) and in the Discussion, paragraph 2 lines 296-300.

Reviewer #2:

1. "The authors convincingly demonstrate a locomotion phenotype in Bsx conditional mouse mutants but no data are presented that Bsx was indeed deleted from the septum and that the deletion is confined to Zic4 lineage cells. The authors should perform immunofluorescence analysis of Bsx in control and mutant brains to demonstrate in which brain regions Bsx protein expression is lost."

Unfortunately, there are no reliable commercial antibodies against BSX. We purchased two such antibodies but none of them detected BSX on sections. In addition, we used an RNA probe spanning the deleted exon 2 to perform in situ hybridization (ISH) on control and cKO embryos but the probe proved to be too small (~150 bp) to detect any transcripts by fluorescent ISH or colour ISH on control or mutant embryos. In the revised manuscript we now demonstrate Cre deletion of the allele by PCR (see new Suppl figure 4). We can indirectly conclude that Bsx is deleted in the septum and not in the hypothalamus (where a phenotype had been observed in germline Bsx mutant mice in Sakkou et al 2007) because there is clear and consistent loss of glutamatergic neurons in the MS in both male and female Zic4-Cre Bsx cKO mice (Figure 6) whereas NPY expression in the hypothalamus is unaffected in Zic4-Cre Bsx cKO mice (Suppl figure 5). The locomotion phenotype observed in our Zic4-Cre Bsx cKO mice is consistent with well-documented functions of MS glutamatergic neurons in the regulation of locomotion. Sakkou et al had concluded that Bsx in the hypothalamus is required for locomotion, disregarding its expression in the septum. Our data, together with known functions of Glu neurons in the MS, are consistent with our conclusion that the locomotion behaviour is due to functions of Bsx in the MS and not the hypothalamus. Of note, Sakkou et al had partially rescued the obesity phenotype of Leptin mutant mice by simultaneously deleting Bsx. However, the locomotion phenotype caused by the absence of Bsx persisted in these rescued mutants. This led to their conclusion that 'Bsx function is an essential genetic component and independent determinant in the hypothalamic regulation of locomotory behavior'. Our data and the literature point instead to a medial septal role for Bsx in locomotion.

2. "Moreover, to make the author's conclusion stronger that glutamatergic neurons in the septum play a crucial role in locomotion, a better comparison of the locomotion phenotypes in global and conditional Bsx mutants would be very informative."

The role of medial septal glutamatergic neurons in locomotion and hippocampal theta is well documented in the literature (see <http://dx.doi.org/10.1113/jphysiol.2005.089664>, <http://dx.doi.org/10.1523/JNEUROSCI.3663-10.2010>, <http://dx.doi.org/10.1016/j.neuron.2015.05.001> etc) In our current manuscript we are simply linking Bsx in the medial septum with glutamatergic neurons (through our fate-mapping) and demonstrating that when Bsx is deleted in the septum, VG2 cells are reduced, resulting in a locomotion phenotype that is consistent with the literature and the role of these cells in the septum. Unfortunately, we do not have the funding or the manpower to extend our behavioural analysis to the germline Bsx KO.

3. "There are some discrepancies between the co-immunostainings and the quantification presented in Figures 2, 3 and 4. For example, according to the quantification in Figure 3, a certain degree of CR+ neurons

originate from the BsxCre lineage, but the figure shows no overlap. The authors should carefully revise the immunostainings in these three figures to correspond with the quantifications".
We have revised the figures to include more representative images in cases where there was a discrepancy. The figures are now representative of the proportions of cells labelled in each Cre line.

Reviewer #3:

1. "A basic schematic describing the connections from the septal complex to other parts of the brain"

Given that in our manuscript we do not investigate the connections of the septum, we feel that it would not be appropriate to include such diagrams. We described the main connections in the introduction, and we refer to excellent published work regarding connectivity.

2. "An explanation of why the immunomarkers used were chosen. What populations do they label?"

When markers are introduced in the text, we now include a sentence to indicate why they were chosen. E.g. lines 92-94, lines 119-120. Apart from the well-studied MSvDB populations, there is very little known about the function of defined populations of LS neurons. We used markers previously reported to be expressed in these regions. The role of these neurons in septal functions remains to be determined.

3. "Relatedly, clarify upfront that P75NTR neurons are cholinergic"

This has been done (line 110).

4. "Incorporate Supplemental Figure 4C into main figures, either as a summary of the results at the end or at the beginning to describe what the Cre driver lines label, or both if you prefer."

We combined suppl figure 2 and part of suppl fig 4c into one main figure (now Figure 2) to describe and summarise the Cre driver lines. We include the summary of the fate-mapping findings as a new Table 2.

5. "Keep a consistent format for how the driver lines fate map each immunomarker. Figure 2a and b was probably the clearest format (as opposed to 3a-e or 4b-d)."

We modified figures 3 and 4 to make them more like figure 2

6. "The other major issue was to do with behavioral analysis of Zic2-Cre; Bsx cKO. It appears that only male mutants exhibit a behavioral phenotype. This was not commented upon either in the results or the discussion."

We now present data for male and female mice both in terms of quantification of septal neurons (Figure 6) and behaviour (Figure 7).

7. "A question related to this: if one were to deconvolve the histological data from Figure 5c by sex, is there a more pronounced effect on cell loss with male mutants? Regardless, this finding needs to be addressed directly in the manuscript."

This has now been done. Male and female mutant mice display similar phenotypes in terms of loss of Glu neurons although the loss is more pronounced in males (Figure 6c and 6f).

8. "The dot plot onto the septal diagram in Fig 1g: Is this one representative slice? Would it be more informative to compile all of the cell placements for all experiments to get a truer sense of cell type placement within lamellae?"

These are representative images at the three different levels and each one has been generated using 2-3 images. Compressing more data into these images will make them too crowded to be of any use.

9. "Figure 1h: indicate the boundary between the MSvDB and the LS"

This has now been done.

10. "Page 4, middle it reads: "The laminar organization of the MS is rearranged at the ventral tier of the vDB". It is unclear what rearranged pertains to."

This has been rectified. The sentence was not necessary and has been removed.

11. "Page 7, second paragraph it reads: "BSX is an evolutionary conserved homeobox-encoding gene..." It should be evolutionarily."

This has been corrected, line 221.

12. "Figure 5c, d: Clarify in the Results text and in the Figure Legend that levels 1-5 indicate anterior to posterior."

This has now been made clear. Lines 238-239.

REVIEWERS' COMMENTS:

Reviewer #1 (Remarks to the Author):

The authors have done an excellent job in responding to most of the questions raised in the previous round of review. I am fully satisfied with this revised version of the manuscript and I recommend it for publication.

Reviewer #2 (Remarks to the Author):

The authors have addressed the issues I raised in my first review. The quality of the manuscript has improved considerably and I recommend publication in Communications Biology.